# Full-Length Genomic RNA of Foot-and-Mouth Disease Virus Is Infectious for Cattle by Injection

**DOI:** 10.3390/v14091924

**Published:** 2022-08-30

**Authors:** Hanna Keck, Benedikt Litz, Bernd Hoffmann, Julia Sehl-Ewert, Martin Beer, Michael Eschbaumer

**Affiliations:** 1Institute of Diagnostic Virology, Friedrich-Loeffler-Institut, Federal Research Institute for Animal Health, Suedufer 10, 17493 Greifswald-Insel Riems, Germany; 2Laboratory for Pathology II, Department of Experimental Animal Facilities and Biorisk Management, Friedrich-Loeffler-Institut, Federal Research Institute for Animal Health, Suedufer 10, 17493 Greifswald-Insel Riems, Germany

**Keywords:** foot-and-mouth disease virus, safe sample transport, single-stranded positive-sense RNA, TRIzol extraction, naked RNA, infectivity, RNA transfection, lipofectamine, self-transfection, BHK cells

## Abstract

Safe sample transport is of great importance for infectious diseases diagnostics. Various treatments and buffers are used to inactivate pathogens in diagnostic samples. At the same time, adequate sample preservation, particularly of nucleic acids, is essential to allow an accurate laboratory diagnosis. For viruses with single-stranded RNA genomes of positive polarity, such as foot-and-mouth disease virus (FMDV), however, naked full-length viral RNA can itself be infectious. In order to assess the risk of infection from inactivated FMDV samples, two animal experiments were performed. In the first trial, six cattle were injected with FMDV RNA (isolate A_22_/IRQ/24/64) into the tongue epithelium. All animals developed clinical disease within two days and FMDV was reisolated from serum and saliva samples. In the second trial, another group of six cattle was exposed to FMDV RNA by instilling it on the tongue and spraying it into the nose. The animals were observed for 10 days after exposure. All animals remained clinically unremarkable and virus isolation as well as FMDV genome detection in serum and saliva were negative. No transfection reagent was used for any of the animal inoculations. In conclusion, cattle can be infected by injection with naked FMDV RNA, but not by non-invasive exposure to the RNA. Inactivated FMDV samples that contain full-length viral RNA carry only a negligible risk of infecting animals.

## 1. Introduction

Diagnostic samples can contain large amounts of infectious disease agents and can be a source of contagion. Inactivating agents or treatments can be used at the point of care to mitigate the risks associated with these samples without compromising their diagnostic utility. For many infectious diseases, the target of first-line diagnostic tests is pathogen-specific nucleic acid. Therefore, inactivating agents such as alcohols or acids, which denature proteins and dissolve lipids, but do not destroy nucleic acids, are often preferred.

Acid treatment is particularly suitable for foot-and-mouth disease virus (FMDV) [1], a high-consequence agricultural pathogen that causes vesicular disease in domestic and wild ruminants and swine. FMDV is an aphthovirus of the *Picornaviridae* family. Like all picornaviruses, FMDV is a non-enveloped virus with a non-segmented single-stranded RNA genome of positive polarity [2]. Unlike cellular mRNA, the viral RNA is not capped, but contains an internal ribosome entry site that allows its direct translation in host cells [2]. Therefore, purified FMDV RNA can initiate an infection in the absence of viral proteins, but in the natural infectious cycle the viral capsid is required for adhesion to the host cell and insertion of the RNA into the cytoplasm [3].

The FMDV capsid readily disassembles at low pH, leading to a rapid loss of infectivity [1]. Accordingly, citric acid is highly effective at inactivating FMDV in tissue samples [4] and on lateral flow devices (LFDs) [5] used for pen-side diagnosis of FMD. Acid-treated samples and LFDs can then be submitted for laboratory analysis under less strict biosafety conditions [6]. Nevertheless, both studies confirmed that these specimens still contain full-length viral RNA, from which infectious virus can be recovered by chemical transfection of permissive cell cultures. It is unknown, however, if animals that come in contact with the full-length viral RNA—accidentally or deliberately—are similarly at risk of FMDV infection.

Several research groups have demonstrated that animals can be infected by naked single-stranded positive-sense genomic RNA of various viruses when transfection reagent is added to the RNA before inoculation [7,8,9]. For porcine reproductive and respiratory syndrome virus it was shown that a productive infection can also be initiated by the injection of viral RNA alone [10].

To investigate the risk of FMDV infection from inactivated diagnostic samples, in this study cattle were exposed to naked FMD RNA by injection and by non-invasive mucosal deposition.

## 2. Materials and Methods

### 2.1. Cells and Viruses

#### 2.1.1. Cell Culture

Two cell lines were used for the cultivation of FMDV: BHK-21 cells (clone “Tübingen”) (RIE 194, Collection of Cell Lines in Veterinary Medicine, FLI, Greifswald, Germany) grown in minimal essential medium (MEM) with Hank’s and Earles’ salts and non-essential amino acids and LFBK-αVβ6 cells [11] (RIE 1419), for which Dulbecco’s modified Eagle medium (DMEM) was used. Both media were supplemented with 10% fetal bovine serum (FBS) during routine culture.

#### 2.1.2. Susceptibility of Cells to Free Viral RNA

To prevent false-positive results in the virus isolation, it is essential that infection of the cell culture is only caused by intact virus particles and there is no spontaneous uptake of free viral RNA by the cells. Accordingly, both cell lines were tested for their self-transfection ability. Two identical 96-well PCR plates were prepared with 200 µL FMDV suspension (strain A_24_ Cruzeiro, approximately 10^7^ TCID_50_/mL) in each well. The plates were then exposed to lengthwise temperature gradients (i.e., with columns of 8 replicate wells heated to the same temperature) of 55–75 °C and 75–95 °C for 10 min in a thermocycler. The heat-treated samples were brought to room temperature, transferred to a tissue culture plate containing either BHK-21 or LFBK-αVβ6 cells, then incubated for 3 days at 37 °C and 5% CO_2_, and subsequently evaluated for cytopathic effect (CPE). A second experiment was completed in the same way, but half of the replicates at each temperature were treated with 5µL RNase A (200 µL/mL) after heating and before cell culture inoculation (BHK-21 cells only).

#### 2.1.3. Selection of Virus Isolates

Three isolates of FMDV, namely A/IRN/8/2015 (lineage A/ASIA/G-VII), A_22_/IRQ/24/64 (also known as “A_22_ Iraq”, lineage A/ASIA/A_22_), and O/FRA/1/2001 (lineage O/ME-SA/PanAsia), were propagated on BHK-21 cells. The virus preparations were analyzed for infectivity by titration on LFBK-αVβ6 cells and for FMDV genome content by RT-qPCR. To test their transfectability, the virus preparations were serially diluted 10-fold in serum-free DMEM down to a final dilution of 10^−5^. Total RNA was extracted from the dilutions and the extracted RNA was analyzed by RT-qPCR and transfection as described below.

#### 2.1.4. Preparation of Virus Stocks

Master stocks of FMDV A_22_/IRQ/24/64 were prepared in BHK-21 cells grown in MEM supplemented with 5% FBS. In a 75 cm^2^ cell culture flask with 16 mL culture medium, 50 µL of a previous preparation of A_22_/IRQ/24/64 were used to inoculate a 90% confluent cell monolayer. About 80% CPE was detected after one day of incubation at 37 °C. After freezing and thawing, the virus suspension was clarified by centrifugation at 1000× *g* for 5 min, aliquoted and stored at −80 °C. Virus and viral RNA content were quantified by end-point titration on LFBK-αVβ6 cells and RT-qPCR.

#### 2.1.5. RNA Extraction and In Vitro Transfection

RNA was extracted from cell culture supernatants using TRIzol LS Reagent (Thermo Fisher Scientific, Waltham, MA, USA) as directed by the manufacturer. The transfections for the preliminary tests and the positive control were performed with Lipofectamine 3000 (Thermo Fisher Scientific). Per well, 1 µL of Lipofectamine 3000 reagent was diluted in 25 µL of DMEM and mixed gently without vortexing. In parallel, 3 µL of extracted RNA, 0.5 µL of P 3000 Reagent, and 25 µL of DMEM were mixed the same way. The two preparations were combined and incubated for 10 min at room temperature, before 50 µL per well were added dropwise to a 24-well plate with near-confluent LFBK-αVβ6 cells. Care was taken not to touch the cell monolayer. After 15 min of incubation at 37 °C, 500 µL of DMEM supplemented with 5% FBS were added per well. CPE was evaluated after 2 days at 37 °C with 5% CO_2_.

#### 2.1.6. RT-qPCR

FMDV RNA was quantified with an RT-qPCR assay targeting the 3D coding region [12], using AgPath-ID One-Step RT-PCR Reagents (Thermo Fisher Scientific) with 2.5 µL of template in a total reaction volume of 12.5 µL.

### 2.2. Material for Inoculation

#### 2.2.1. Assessment of RNA Degradation by Atomization

Before the animal experiment, it was tested if nebulization of FMDV RNA leads to significant fragmentation, as had been observed for large DNA molecules [13]. Half of the pooled RNA extracted from an FMDV A_22_/IRQ/24/64 culture was serially diluted in nuclease-free water and transfected into LFBK-α_V_β_6_ cells using two replicate wells per dilution. The other half was nebulized using a syringe-mounted atomization device (MAD Nasal, Teleflex, Morrisville, NC, USA). Nebulization was performed in a biological safety cabinet and the nebulized RNA was collected in a 50 mL conical tube for dilution and transfection. After 2 days of incubation at 37 °C with 5% CO_2_, the transfected cells were examined for CPE.

#### 2.2.2. Preparation of RNA Inoculum

To prevent the viral RNA from being damaged by freezing and thawing and obtain as many intact full-length viral genomes as possible, it was extracted immediately before inoculation of the animals in each experiment. For the first experiment, the pooled RNA of 18 replicate extractions was diluted 1:5 in nuclease-free water. For the second experiment, the pooled RNA was also diluted 1:5, but half of the 1:5 dilution was then further diluted to provide an adequate volume for intranasal administration by atomizer. For this purpose, 1.2 mL of the diluted pooled RNA were added to 11.4 mL of nuclease-free water. The freshly extracted and diluted RNA was transported to the animal room on ice. No transfection reagent was added to the viral RNA for any of the animal inoculations.

The leftover RNA inoculum was brought back to the laboratory on wet ice for confirmatory tests.

#### 2.2.3. Negative Control: Exclusion of Virus Contamination

To confirm that it did not contain any intact viral particles, 50 µL of the RNA inoculum without transfection reagent were added to an intact monolayer of LFBK-αVβ6 cells and incubated at 37 °C for 3 days. The test was carried out in three replicates. After 3 days, the cultures were frozen and thawed and the cell lysate was passaged onto fresh LFBK-αVβ6 cells. After another 3 days of incubation, the cultures were examined for CPE.

#### 2.2.4. Positive Control: Demonstration of Full-Length Viral RNA

To confirm that intact full-length FMDV genomes were present in the inoculum, it was serially diluted in nuclease-free water and transfected into LFBK-αVβ6 cells using lipofectamine as described above. The viral RNA content of the inoculum was quantified with the FMDV 3D RT-qPCR assay. A serial dilution of an in vitro transcribed FMDV RNA standard was included to construct a calibration curve and calculate genome copy numbers.

### 2.3. Animal Experiments

#### 2.3.1. Animals and Ethics Statement

Six Holstein-Friesian cattle were used for each trial. The animals were brought to the BSL4vet facility of the FLI Riems one week before the start of the trial to ensure proper acclimation. In the first experiment, the animals were between 5 and 8 months of age. In the second experiment, they were between 3 and 4 months.

For the inoculations and clinical exams, the animals were kept off feed overnight and sedated by intramuscular injection of 0.3 mg xylazine per kg body mass into the hind quarters. The sedation was reversed by intramuscular injection of 0.03 mg atipamezole per kg body mass.

All work with animals occurred after ethical review and in compliance with local, state, and national animal welfare regulations. The experimental protocol was filed with the State Office for Agriculture, Food and Fisheries of Mecklenburg-Vorpommern (LALLF M-V), the competent authority for animal experiments conducted at FLI Riems (file no. 7221.3-2-82-026/17). The animals were handled in accordance with all applicable European and German guidelines for the use of experimental animals.

#### 2.3.2. Experimental Design

The first animal experiment was designed to clarify whether it is at all possible to cause an FMDV infection in cattle with naked RNA. Accordingly, the most reliable route of inoculation was used: injection into the epithelium of the tongue [14]. The tongue of the deeply sedated animals (ear tag numbers 248, 249, 251, 260, 262, and 263) was pulled out of the mouth and the 1:5 diluted RNA was injected as superficially as possible with a fine hypodermic needle until a small bleb was formed around the tip of the needle. Four injections were made and 100 µL of RNA was injected at each site, but it is not possible to say with certainty how much of the inoculum remained in the tongue epithelium [15].

In the second experiment, less invasive routes of inoculation were used. Half of the 1:5 diluted RNA (200 µL per animal) was dripped onto the tongue of the sedated animals. The tongues were inspected before application of the RNA to make sure that there were no visible breaks in the epithelium. For each animal, 2.1 mL of the further diluted RNA preparation was applied to the nasal mucosa by spraying into a nostril using a syringe-mounted atomizer (MAD Nasal, Teleflex). The stepwise application of the total volume of RNA was timed to coincide with inspiration.

#### 2.3.3. Monitoring and Sample Collection during the Trials

Rectal body temperature was documented daily during the acclimation and experimental period. After inoculation, the animals were examined each day for signs of FMD. A clinical score was calculated by evaluation of the general attitude, feed intake, body temperature, and gait (see Appendix A for details on the evaluation criteria).

Serum and saliva samples were collected immediately before inoculation and daily thereafter. An additional saliva sample was collected immediately after inoculation. Blood was drawn from the jugular vein into collection tubes with clotting activators (Kabevette, Kabe Labortechnik, Nümbrecht, Germany). Saliva was collected by swabbing the oral cavity with a human vaginal tampon (o.b., Johnson & Johnson, Neuss, Germany) to quickly obtain a large quantity of fluid. Samples were transferred back to the laboratory on wet ice, centrifuged immediately for 10 min at 2000× *g* and 4 °C, then aliquoted and stored at −80 °C until further analysis.

#### 2.3.4. Necropsy

At the end of the trial or when a humane endpoint was reached, the animals were deeply sedated as described above and then euthanized by intravenous injection of an overdose of sodium pentobarbital (Release, WdT eG, Garbsen, Germany). Samples of the lesions on the tongues and in the interdigital spaces were taken at necropsy and placed in 10% formalin for histopathological examination.

For determination of the viral RNA content, small pieces of vesicular epithelium (2 mm × 2 mm) from the same sites were collected in a screw-cap tube without fixative.

### 2.4. Sample Processing

#### 2.4.1. Virus Isolation and RT-qPCR in Ex Vivo Samples

The collected serum and saliva samples were checked for the presence of infectious virus by isolation on LFBK-αVβ6 cells. For this purpose, 12.5 cm^2^ tissue culture flasks with 90% confluent cell monolayers were inoculated with 50 µL of sample material. The assay was performed in duplicate. After incubation for 3 days at 37 °C, the flasks were evaluated for CPE and frozen at −80 °C to disrupt the cells. After thawing, 1 mL of the lysate was passaged onto fresh LFBK-αVβ6 monolayers, incubated, and again evaluated for CPE.

The tissue samples collected for RT-qPCR were homogenized in 500 µL serum-free DMEM with a 4 mm stainless steel bead at 30 shakes/s for 3 min in a Tissue Lyser II (Qiagen, Venlo, The Netherlands). The homogenate was centrifuged for 2 min at 20,000× *g* at 4 °C, the supernatant was collected and used for RNA extraction with the QIAamp Viral RNA Kit (Qiagen).

The viral RNA content of the serum and saliva samples, as well as of vesicular fluid collected from lesions and homogenized vesicular epithelium, was quantified with the 3D RT-qPCR assay.

#### 2.4.2. Sequence Analysis

The serum samples of all animals from day 2 of the first experiment were used for FMDV sequence analysis. RNA was extracted using the QIAamp Viral RNA Kit (Qiagen). The VP1 coding region was amplified using the previously published primers FMD-3161-F and FMD-4303-R [16] with the qScript XLT One-Step RT-qPCR ToughMix (Quanta, Beverly, MA, USA). The PCR product was purified by agarose-gel electrophoresis. Bands of the expected size (1.2 kbp) were excised, the DNA was extracted and sent to Eurofins (Ebersberg, Germany) for Sanger sequencing.

#### 2.4.3. Histopathology

Histological sections of all tissue samples taken during the first animal experiment were prepared. These were each stained with hematoxylin and eosin (H.E.) as well as with an in-house polyclonal rabbit anti-FMDV antiserum (1:2000). Adjacent tissue sections treated with pre-immune serum served as a negative control.

Since there were no FMDV-positive ex vivo samples and no conspicuous lesions in the second animal experiment, no histological examination was performed.

### 2.5. Statistical Analysis

For the statistical analysis, the animal inoculations were considered Bernoulli trials with a binary outcome (infection/no infection). Accordingly, each animal experiment comprised six independent observations, from which 95% confidence intervals for the proportion of infection were calculated using Jeffreys’ method as implemented in the Epitools suite (https://epitools.ausvet.com.au/ciproportion) (accessed on 21 July 2022).

## 3. Results

### 3.1. Susceptibility of Cells to Free FMDV RNA

Without RNase treatment of the heated virus suspension, there was a clear difference between LFBK-αVβ6 and BHK-21 cells. After heating to about 60 °C for 10 min, the virus suspension no longer caused infection in LFBK-αVβ6 cells. In BHK-21 cells, on the other hand, CPE did still occur up to a temperature of about 80 °C (Figure 1). When RNase A was added to the heat-treated virus suspension before inoculation of BHK-21 cells, there no longer was any difference between the cell lines and no CPE was seen for any sample heated to at least 60 °C for 10 min.

### 3.2. Virus Stocks for Animal Experiments

The preparations of FMDV A/IRN/8/2015, A_22_/IRQ/24/64, and O/FRA/1/2001 had virus titers of 10^7.3^, 10^8.4^, and 10^5.3^ TCID_50_/mL with C_t_ values of 12.3, 12.5, and 14.4, respectively. The transfection was successful for O/FRA/1/2001 down to a dilution of 10^−2^, for A/IRN/8/2015 to 10^−3^, and for A_22_/IRQ/24/64 to a dilution of 10^−4^, which corresponded to a C_t_ value of 26.5. Based on these results, A_22_/IRQ/24/64 was selected for the animal experiments, because it provided the largest margin of safety for successful transfection.

The master stock of A_22_/IRQ/24/64 used in the second animal trial had a virus titer of 10^7.8^ TCID_50_/mL with a C_t_ value of 14.4.

### 3.3. Assessment of RNA Degradation by Atomization

The comparison of untreated and previously nebulized RNA showed no difference in transfectability. For both preparations, transfection was successful in at least one of two replicates down to a dilution of 10^−4^.

### 3.4. Confirmatory Tests of RNA Inoculum

The RNA preparations used to inoculate the animals had FMDV C_t_ values of 13.7 in the first experiment and 13.3 in the second, corresponding to about 10^11^ copies of FMDV genome in each animal dose.

Cultures of LFBK-αVβ6 cells inoculated with the RNA preparations without transfection reagent did not exhibit any CPE in either the first or second animal experiment.

With lipofectamine, the RNA inoculum from the first experiment was successfully transfected into LFBK-αVβ6 down to a dilution of 10^−3^. For the second experiment, in vitro transfection of the inoculum was successful down to a dilution of 10^−4^.

### 3.5. Clinical Findings

In the first animal experiment, excessive salivation was observed in all animals 24 h after inoculation. In addition, two animals exhibited an increased body temperature of more than 39.5 °C (see Figure 2). Cursory examination of the oral cavity without sedation revealed conspicuously raised areas on the tongue epithelium of 4 of the 6 animals. The following day, 48 h after inoculation, all 6 animals were sedated for a comprehensive clinical exam. Clearly visible lesions were found on the tongues of all animals, corresponding to the areas of injection, but also on the upper gums. Vesicles, some of which were still intact, were found in the interdigital spaces of all animals. One animal (no. 248) was euthanized immediately, while the remaining 5 cattle received anti-inflammatory analgesics and necropsy was scheduled for the next day.

In contrast, no clinical signs of FMD were seen in any of the 6 cattle in the second animal experiment.

### 3.6. Virus Isolation

All samples taken on days 1 and 2 after inoculation in the first animal experiment were positive in the virus isolation. No positive samples were found in the second experiment.

### 3.7. FMDV RT-qPCR

No FMDV genome was detected in the serum and saliva samples collected before infection in the first animal experiment. Beginning on the first day after inoculation, however, a steady increase in the viral genome load was observed in serum and saliva (see Figure 3). The homogenized vesicular material of the tongue lesions had an average C_q_ value of 16.6 in the FMDV RT-qPCR.

In the second animal experiment, no FMDV genome was detected in any sample.

### 3.8. Sequencing

The viral sequence recovered from the samples of the first animal experiment matched the database sequence of FMDV A_22_/IRQ/24/64 to at least 99.7%.

### 3.9. Necropsy

The clinical findings of the first animal trial were confirmed during necropsy (see Figure 4). The tongues of all animals showed extensive, mostly superficial, but occasionally deeper lesions of the epithelium. Lesions were also found on the lip, gums, and hard palate as well as in the interdigital spaces. Several vesicles in the interdigital space were still intact at the time of necropsy and vesicular fluid was recovered.

No lesions of this kind were found in the second animal experiment.

### 3.10. Histology

Vesiculoulcerative tissue damage can be seen in the H.E. staining in all samples of the first animal experiment (Figure 5). This coincides with specific staining that indicates the presence of FMDV antigen in the cells.

### 3.11. Statistical Analysis

In the first experiment, the proportion of animals in which the injection of FMDV RNA into the tongue led to infection was 100%, with a lower bound of the 95% confidence interval of 67%.

In the second animal experiment, none of the animals developed FMD, corresponding to an observed proportion of infection after non-invasive exposure to FMDV RNA of 0%. The upper bound of the 95% confidence interval was 33%.

## 4. Discussion

### 4.1. Self-Transfection of BHK-21 Cells

The critical temperature for FMDV inactivation is about 60 °C [1]. Above this threshold, the viral capsid is denatured and infectivity is abolished. This is in contrast with the observation that BHK-21 cells can become infected by virus suspensions that were heated to 80 °C. Since this was prevented by the addition of RNase A, we assume that the infection is caused by the viral RNA that is released when the capsid is destroyed. BHK-21 cells appear to take up RNA spontaneously from the surrounding medium. In a cell culture environment, optimal conditions for the cells in terms of nutrients, temperature, partial CO_2_ pressure, and ambient pH prevail, and the cellular membrane of BHK-21 cells may have become increasingly permeable over the course of many passages. This can be beneficial in certain applications, e.g., for the rescue of infectious virus in reverse genetics.

However, the potential for “self-transfection” must be kept in mind when setting up virological assays. When BHK-21 cells are used for virus titrations, CPE is not only induced by intact virions but also by naked RNA and it is conceivable that slightly higher titer readings are obtained compared to other cells. Self-transfection can also lead to divergent results in inactivation tests of FMDV, which are carried out to assess the degree of stability of the virus particle when exposed to chemicals, heat, or radiation. The treated virus preparation is inoculated into cell culture and the cultures are evaluated for the development of CPE [1,17,18,19,20]. When no CPE develops after two or more passages, the sample is considered sufficiently inactivated. If the used method of inactivation does not sufficiently degrade the viral genome, however, using BHK-21 cells for the test readout can underestimate the extent of virus inactivation.

We did not see any indication of self-transfection with LFBK-αVβ6 cells. Based on this observation and on their high sensitivity for FMDV infection, we used these cells exclusively for all tests of the RNA inoculum and the samples from the animal experiments.

### 4.2. Infectivity of Full-Length FMDV RNA

The first experiment clearly demonstrated that cattle can be infected with FMDV by a superficial injection of naked viral RNA into the tongue. The occurrence of lesions in the interdigital space indicates that the infection became generalized within less than two days. The calculated inoculation dose of 10^11^ FMDV genomes is an upper bound, because the RT-qPCR assay does not distinguish between whole genomes and viral RNA fragments, as long as they contain the part of the 3D-coding region targeted by the PCR.

Certainly, the large amount of viral genome from a preparation selected for high transfectability created optimal conditions for successful infection. The rapid course of disease—no slower than after inoculation of infectious FMDV [21]—suggests that a significantly lower amount of injected RNA could also cause infection, but we did not attempt to determine a minimal infectious dose. This would be of limited use, since variables such as the quality of the RNA preparation, the proportion of full-length genomes in the total viral RNA, the choice of FMDV strain, the number of inoculation sites, the anatomic location of the inoculation, and the route of application will also have a major influence on the course of infection.

When conceiving of this study, we were not aware of any prior work where FMDV RNA was injected into animals without a transfection reagent. It has since been brought to our attention that there was in fact such a study in the 1950s [22], which also found that naked RNA extracted from vesicular epithelium of cattle infected with FMDV O_2_ Spain, A_5_ Eystrup, or C_1_ Tölz was infectious for cattle and guinea pigs.

The non-invasive application of FMDV RNA in the second experiment, i.e., instillation on the tongue and intranasal nebulization, did not result in infection despite the very high amount of viral RNA. This is probably because an intact cellular membrane of epithelial cells efficiently prevents the entry of naked RNA under natural conditions. Saliva also contains a large number of different ribonucleases [23,24]. These play an important role in host defense [25] due to their antiviral properties, including [26] but not limited to the degradation of free RNA [27,28].

However, as soon as the barrier of at least one cell is breached, e.g., if it is nicked by a hypodermic needle, a single intact full genome that is introduced to the cytoplasm can cause a productive FMDV infection. For this to occur, it is important that the damage is just large enough for the RNA to gain entry to the cell, but not so large that the cell dies. While a similar injury can be caused by chewing on a jagged piece of plastic from a damaged LFD, it must be emphasized that this scenario is highly contrived and seems very unlikely.

The main risk of spreading FMDV over longer distances certainly lies elsewhere, such as in trade with infected animals or contaminated materials [29]. Based on our results, it can be concluded that the risk of FMDV transmission through the shipment of properly inactivated samples is negligible and it is reasonable to treat these samples as exempt from most dangerous goods regulations. However, to further reduce the risk of deliberate diversion of the material or its inadvertent release to the environment, such shipments should only be handled by reliable carriers and should be tracked closely while in transit [6].

## Figures and Tables

**Figure 1 viruses-14-01924-f001:**
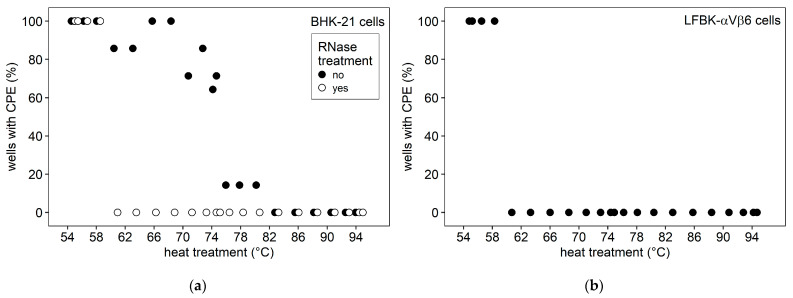
Inoculation of cell cultures with heat-treated FMDV. For each treatment temperature, the percentage of replicate wells that developed CPE 72 h after inoculation with the previously heated virus suspension is shown: (**a**) heat-inactivated virus preparation on BHK-21 cells, with and without RNase treatment after heating; (**b**) heat-inactivated virus preparation on LFBK-αVβ6 cells.

**Figure 2 viruses-14-01924-f002:**
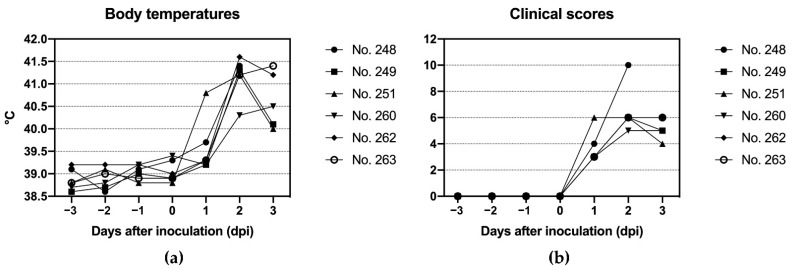
Clinical findings in the first animal experiment after injection of naked FMDV RNA into the tongue: (**a**) rectal body temperatures; (**b**) clinical scores. See Appendix A for the raw data summarized in this figure.

**Figure 3 viruses-14-01924-f003:**
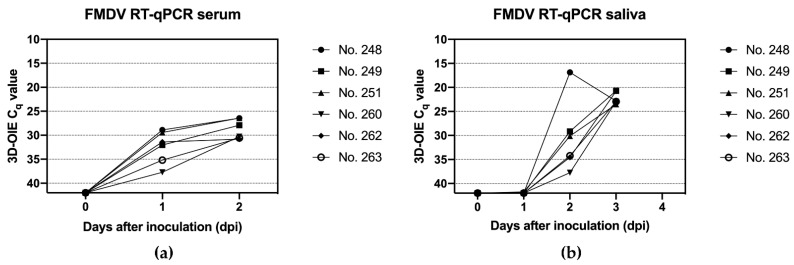
FMDV RT-q-PCR results after injection of naked FMDV RNA in the first animal trial: (**a**) saliva samples; (**b**) serum samples. See Appendix A for the raw data summarized in this figure.

**Figure 4 viruses-14-01924-f004:**
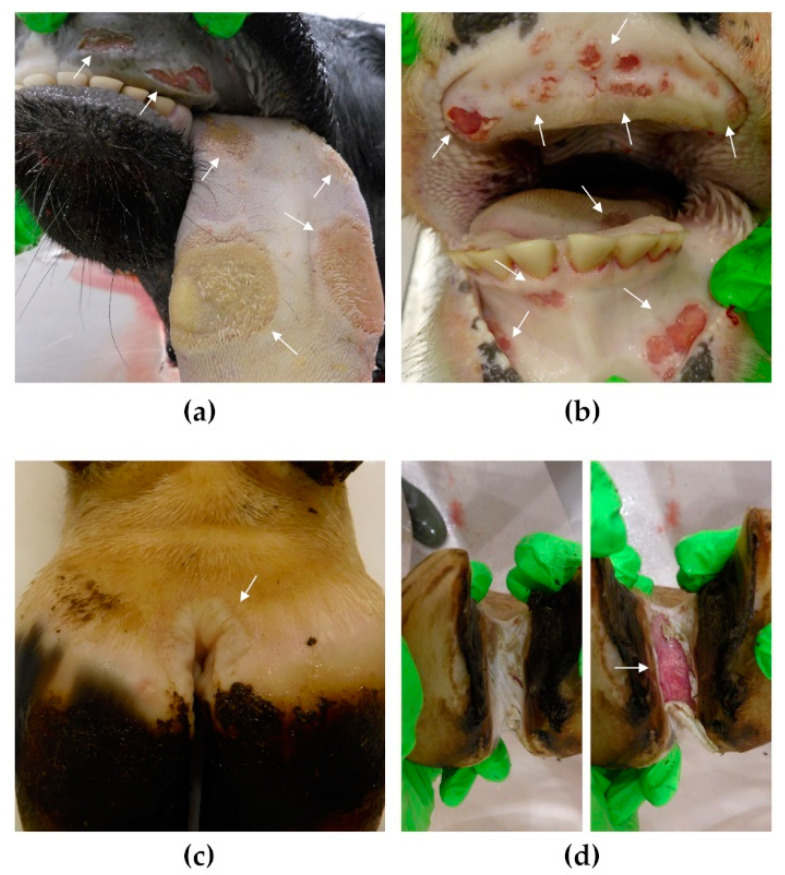
Postmortem findings after injection of naked FMDV RNA into the tongue. All six cattle in the first animal trial developed severe clinical signs of FMD and pathognomonic findings were made at necropsy (arrows): (**a**) lesions on the tongue, corresponding to the areas of injection, and on the upper gums; (**b**) lesions on the tongue, the upper and lower gums; (**c**) large, mostly intact lesion in the interdigital space; (**d**) the true extent of the lesion is revealed after manipulation.

**Figure 5 viruses-14-01924-f005:**
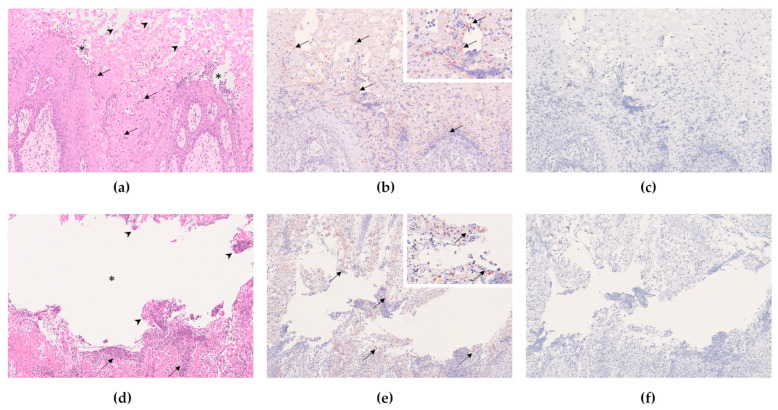
Pathohistological findings after injection of naked FMDV RNA into the tongue in the first animal trial ((**a**–**c**), tongue, 5× magnification; d–f, interdigital space, 10× magnification). (**a**) Animal 249, tongue, H.E. staining. The lingual mucosa shows characteristic vesiculopustular lesions containing neutrophils and sloughed epithelial cells (asterisk). The adjacent mucosa is infiltrated by neutrophils (arrows) and reveals clear intercellular spaces between swollen and degenerating keratinocytes (arrowheads). (**b**) Animal 249, tongue, anti-FMDV immunohistochemistry. Associated with the lesions as described in a), FMDV-antigen-positive cells (red–brown signal, arrows) were detected throughout the lingual mucosal epithelium. The inset shows positively stained cells at higher magnification. (**c**) Animal 249, tongue, pre-immune serum immunohistochemistry (negative control). To confirm the specificity of the anti-FMDV antibody staining, tissue sections were incubated with pre-immune serum. Unspecific binding was excluded as no positively labeled cells were detected. (**d**) Animal 260, interdigital space, H.E. staining. Overview of a large vesicle in the interdigital cleft. A large intraepidermal space (asterisk) is filled with neutrophils and degenerate keratinocytes (arrowheads). The adjacent epidermis is severely infiltrated by neutrophils (arrows) intermingled with degenerating keratinocytes. (**e**) Animal 260, interdigital space, anti-FMDV immunohistochemistry. Numerous cells within the lesion were FMDV-antigen positive (arrow). The inset shows positively stained cells at higher magnification. (**f**) Animal 260, interdigital space, pre-immune serum immunohistochemistry. As described for (**c**), unspecific binding of the FMDV antibodies was ruled out as no positive signal was detectable with the pre-immune serum.

## Data Availability

All quantitative data from the study are available in the Appendix A.

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
