# Peer review of "Full-Length Genomic RNA of Foot-and-Mouth Disease Virus Is Infectious for Cattle by Injection"

_viruses, 2022, doi:10.3390/v14091924_

Round 1
Reviewer 1 Report
August 9th, 2022
Review: viruses-1853581
“Full-Length Genomic RNA of Foot-and-Mouth Disease Virus is Infectious for Cattle by Injection”
In this manuscript the authors describe the infectivity of FMDV full-length naked RNA when injected parenterally into cattle tongue epithelium but not when the inoculation route chosen for exposure is a more natural one, instilling on the tongue and spraying on the nose. The topic is of great interest to understand what is the importance of sample preservation in FMD-infected samples and better adapt decontamination protocols. However, there are important pieces of information that are not presented clearly in the manuscript. For instance, there is not a clear normalization of the amount of RNA inoculated, nor some data of the quality of the RNA analyzed by a bioanalyzer or tape-station.
Minor comments:
1. Why didn’t the authors choose to use a well characterized non-invasive, more natural, route of FMDV inoculation, like the intranasopharyngeal route?
2. It is my understanding that it is not entirely correct to use “skin” to refer to tongue epithelium, since the anatomy of the tissue in the tongue is not the same as the skin. Please, revise the text and change to tongue epithelium when needed.
3. Would it be of interest to learn what is the threshold amount of RNA sufficient to get animals infected?
Author Response
Reviewer 1
In this manuscript the authors describe the infectivity of FMDV full-length naked RNA when injected parenterally into cattle tongue epithelium but not when the inoculation route chosen for exposure is a more natural one, instilling on the tongue and spraying on the nose. The topic is of great interest to understand what is the importance of sample preservation in FMD-infected samples and better adapt decontamination protocols.
Response: We thank the reviewer for their positive assessment of the relevance of the topic!
However, there are important pieces of information that are not presented clearly in the manuscript. For instance, there is not a clear normalization of the amount of RNA inoculated, nor some data of the quality of the RNA analyzed by a bioanalyzer or tape-station.
Response: The number of viral genomes in the inoculated RNA preparations was quantified by real-time RT-PCR. A description of the quantification method was added to the manuscript as requested by Reviewer 2. No further analysis of the RNA was undertaken because we did not expect a useful result. The only RNA analysis method available in the FMD lab itself is a NanoDrop spectrophotometer. The bulk of the extracted RNA will be of cellular origin and any measurement would primarily report on the cellular RNA. In order to use a Bioanalyzer or TapeStation, the RNA has to be heated or chemically treated to move it out of the FMD containment. Any measurement subsequently obtained from the treated RNA would not be applicable to the original preparation that was inoculated into the animals.
Minor comments:
- Why didn’t the authors choose to use a well characterized non-invasive, more natural, route of FMDV inoculation, like the intranasopharyngeal route?
Response: We are proponents of the intranasopharyngeal (INP) route of application ourselves and have used it in several studies. However, in this study, we intended to recreate a scenario where material containing full-length FMDV RNA (e.g., a tube with an inactivated diagnostic sample or an acid-treated lateral flow device) is accidentally consumed by an animal. Therefore, the RNA was deposited in the mouth and in the anterior nasal cavity rather than in the nasopharynx.
- It is my understanding that it is not entirely correct to use “skin” to refer to tongue epithelium, since the anatomy of the tissue in the tongue is not the same as the skin. Please, revise the text and change to tongue epithelium when needed.
Response: This has been changed as requested by the reviewer (lines 179, 183, 288). Line numbers refer to the final version of the revised manuscript without tracked changes.
- Would it be of interest to learn what is the threshold amount of RNA sufficient to get animals infected?
Response: Scientifically, it would be interesting to determine a minimal infectious dose of full-length RNA, possibly in comparison to intact viral particles applied in the same way. However, this would be of limited practical use, since variables such as the quality of the RNA preparation, the proportion of full-length genomes in the total viral RNA, the choice of FMDV strain, the number of inoculation sites, the anatomic location of the inoculation, and the route of application will also have a major influence on the course of infection.
Reviewer 2 Report
The proposed paper aims to provide new information, useful to determine the biological safety of the transport of inactivate full-length genome of FMD viruses. In detail, it experimentally analyses the infectivity of the RNA genome when (i) superficially injected and (ii) instilled on the tongue or intranasally nebulized. Moreover, it clearly suggests the possibility that some susceptible cell-line react to naked viral genome, self-transfecting the RNA without transfection reagents.
The importance of the subject is great because it is still a current and much debated topic, and the impact of the obtained data is clearly big.
Following, the authors can find some criticisms I found in their manuscript. Unfortunately, in some cases it is not easy to explain to which point of the draft I am referring, because the layout of the text does not include numbers for the lines. My apologise.
1. Some points of the introduction is poor of bibliography notes. In detail, I would appreciate notes confirming i) that “acid treatment is particularly suitable for… FMDV”, ii) that “FMDV is a … single-stranded RNA genome of positive polarity” and iii) that “viral capsid is required for adhesion to the host cell and insertion of the RNA into the cytoplasm”.
2. In the paragraph 3.2 you described the preparations of virus stocks and the dilutions used for effective transfections. A IRN 15 had the titre 107.3 TCID50/ml and diluted 10-3 induced a successful transfection (minimum transfecting titre 104.3 TCID50/ml); A22 had the titre 108.4 TCID50/ml and diluted 10-4 induced a successful transfection (minimum transfecting titre 104.4 TCID50/ml); O FRA 01 had the titre 105.3 TCID50/ml and diluted 10-2 induced a successful transfection (minimum transfecting titre 103.3 TCID50/ml). If I am not wrong, type O showed the best transfection fitness because the lowest titre of virus type O was needed to transfect (1 log lower than the type A). Nevertheless, it is clear that the richest virus stock was the A22 (108.4 TCID50/ml) and was the one which could be more diluted for the following experiments (starting from 16 ml of suspension you could obtain 160 L of transfection-effective virus suspension). In the text you wrote “Based on these results…” but you do not clearly specify the factor you followed for the decision of the strain to use. I guess you considered the “economy” factor instead of the efficacy of the transfection, but it would be indicated unquestionably.
3. In the paragraph 3.4 you stated that “…the RNA preparations used to inoculate… corresponding to about 1011 copies…”. In the M&M section you cited an RT-qPCR, but in my opinion, it would be better to add how you quantified the genome copies (did you produce a calibration curve or is it just an approximated evaluation?).
4. In the paragraph 3.5 you asserted “…a slight increase in body temperature was observed in all animals 24 hours after inoculation…”. In my opinion, this is not accurate. At 0 dpi the body temperature was between 38.8 and 39.4 degrees with a mean of 39.05. At 1 dpi, excluding two animals, 251 (40.8°C) and 248 (39.7°C), the other four animals (67%) did not show a body temperature increase (temp between 39.2 and 39.3 degrees with a mean of 39.25). Even considering the scores assigned to body temperature, four out of six animals were 0. For these reasons, the original assessment (all animals increased temperature at 1 dpi) needs to be changed.
5. Figure 4. For better clarity, especially for people unfamiliar with FMDV clinical signs or with veterinary diagnosis, I think it would be useful to highlight the lesions with signs (e.g. arrows, arrowheads, asterisks…).
6. Figure 5. As before, I think that it would be better to increase the clarity of the figure, especially for people unfamiliar with the histology of FMDV tissue damages. You could i) clearly indicate the ruptures, the cells/tissue involved in the lesions, the lesions themselves, the cells positive for FMDV antibody; ii) explain better what the pictures are showing (in the caption or in the text); iii) add a good and comparable negative control, because picture c and f could be compared to b and e but not to a and d because of a very different coloration. This figure needs huge rearrangement in order to be more explicative.
Author Response
Reviewer 2
The proposed paper aims to provide new information, useful to determine the biological safety of the transport of inactivate full-length genome of FMD viruses. In detail, it experimentally analyses the infectivity of the RNA genome when (i) superficially injected and (ii) instilled on the tongue or intranasally nebulized. Moreover, it clearly suggests the possibility that some susceptible cell-line react to naked viral genome, self-transfecting the RNA without transfection reagents.
The importance of the subject is great because it is still a current and much debated topic, and the impact of the obtained data is clearly big.
Response: We thank the reviewer for their positive assessment of the relevance of the topic!
Following, the authors can find some criticisms I found in their manuscript. Unfortunately, in some cases it is not easy to explain to which point of the draft I am referring, because the layout of the text does not include numbers for the lines. My apologise.
Response: We apologize for this oversight. Line numbers are now included.
- Some points of the introduction is poor of bibliography notes. In detail, I would appreciate notes confirming i) that “acid treatment is particularly suitable for… FMDV”, ii) that “FMDV is a … single-stranded RNA genome of positive polarity” and iii) that “viral capsid is required for adhesion to the host cell and insertion of the RNA into the cytoplasm”.
Response: References have been added as requested by the reviewer (lines 38, 42, 46).
- In the paragraph 3.2 you described the preparations of virus stocks and the dilutions used for effective transfections. A IRN 15 had the titre 10^7.3 TCID50/ml and diluted 10^-3 induced a successful transfection (minimum transfecting titre 10^4.3 TCID50/ml); A22 had the titre 10^8.4 TCID50/ml and diluted 10^-4 induced a successful transfection (minimum transfecting titre 10^4.4 TCID50/ml); O FRA 01 had the titre 10^5.3 TCID50/ml and diluted 10^-2 induced a successful transfection (minimum transfecting titre 10^3.3 TCID50/ml). If I am not wrong, type O showed the best transfection fitness because the lowest titre of virus type O was needed to transfect (1 log lower than the type A). Nevertheless, it is clear that the richest virus stock was the A22 (10^8.4 TCID50/ml) and was the one which could be more diluted for the following experiments (starting from 16 ml of suspension you could obtain 160 L of transfection-effective virus suspension). In the text you wrote “Based on these results…” but you do not clearly specify the factor you followed for the decision of the strain to use. I guess you considered the “economy” factor instead of the efficacy of the transfection, but it would be indicated unquestionably.
Response: A22 was selected because it provided the largest margin of safety for successful transfection, i.e. the virus stock could be diluted the furthest (10^-4 compared to 10^-3 and 10^-2 for the other viruses) while still giving positive results in the transfection. This is now stated in the manuscript (line 268).
- In the paragraph 3.4 you stated that “…the RNA preparations used to inoculate… corresponding to about 10^11 copies…”. In the M&M section you cited an RT-qPCR, but in my opinion, it would be better to add how you quantified the genome copies (did you produce a calibration curve or is it just an approximated evaluation?).
Response: Quantification was done with a calibration curve. This is now stated in the manuscript (line 154).
- In the paragraph 3.5 you asserted “…a slight increase in body temperature was observed in all animals 24 hours after inoculation…”. In my opinion, this is not accurate. At 0 dpi the body temperature was between 38.8 and 39.4 degrees with a mean of 39.05. At 1 dpi, excluding two animals, 251 (40.8°C) and 248 (39.7°C), the other four animals (67%) did not show a body temperature increase (temp between 39.2 and 39.3 degrees with a mean of 39.25). Even considering the scores assigned to body temperature, four out of six animals were 0. For these reasons, the original assessment (all animals increased temperature at 1 dpi) needs to be changed.
Response: This has been changed as requested by the reviewer (line 286).
- Figure 4. For better clarity, especially for people unfamiliar with FMDV clinical signs or with veterinary diagnosis, I think it would be useful to highlight the lesions with signs (e.g. arrows, arrowheads, asterisks…).
Response: Markings have been added as requested by the reviewer.
- Figure 5. As before, I think that it would be better to increase the clarity of the figure, especially for people unfamiliar with the histology of FMDV tissue damages. You could i) clearly indicate the ruptures, the cells/tissue involved in the lesions, the lesions themselves, the cells positive for FMDV antibody; ii) explain better what the pictures are showing (in the caption or in the text); iii) add a good and comparable negative control, because picture c and f could be compared to b and e but not to a and d because of a very different coloration. This figure needs huge rearrangement in order to be more explicative.
Response: Markings have been added and the figure caption has been expanded as requested by the reviewer.
Reviewer 3 Report
Dear authors
Thanks for this interested manuscript: Full-Length Genomic RNA of Foot-and-Mouth Disease Virus is Infectious for Cattle by Injection
In order to assess the risk of infection from inactivated FMDV samples, two animal experiments was done
Introduction need more on newest paper on FMD as: https://doi.org/10.3390/ani11061697
2.3.4. Necropsy ; change to Histopathology
Author Response
Reviewer 3
Dear authors
Thanks for this interested manuscript: Full-Length Genomic RNA of Foot-and-Mouth Disease Virus is Infectious for Cattle by Injection
Response: We thank the reviewer for the positive assessment of our manuscript!
In order to assess the risk of infection from inactivated FMDV samples, two animal experiments was done
Introduction need more on newest paper on FMD as: https://doi.org/10.3390/ani11061697
Response: The paper suggested by the reviewer, “Characterization of Foot and Mouth Disease Virus Serotype SAT-2 in Swamp Water Buffaloes (Bubalus bubalis) under the Egyptian Smallholder Production System”, has little or no relevance for our study. We do not think it is appropriate to cite this paper in the introduction.
2.3.4. Necropsy; change to Histopathology
Response: The heading is appropriate; section 2.3.4 only describes the necropsy. Histopathology is described in section 2.4.3, whose heading has been changed as requested.
Reviewer 4 Report
This study is very important to know the risk of diagnosis materials for RT-PCR etc. on FMD, because the cattle can be infected by injection with naked FMDV RNA. But it cannot by non-invasive exposure for example intranasal nebulization the full length RNA to cattle. Therefore, it could be concluded that an inactivated FMDV samples that contain full-length viral RNA carry only a negligible risk of infection to animals. However, it is better to add the sentences bellow at last of conclusion in discussion.
Inactivated FMDV samples that contain full-length viral RNA carry only a negligible risk of infecting animals. But cattle can be infected by direct injection with naked FMDV RNA, therefore, it should be remarkable these results when diagnosis of FMD.
Author Response
Reviewer 4
This study is very important to know the risk of diagnosis materials for RT-PCR etc. on FMD, because the cattle can be infected by injection with naked FMDV RNA. But it cannot by non-invasive exposure for example intranasal nebulization the full length RNA to cattle.
Response: We thank the reviewer for their positive assessment of the relevance of the topic!
Therefore, it could be concluded that an inactivated FMDV samples that contain full-length viral RNA carry only a negligible risk of infection to animals. However, it is better to add the sentences bellow at last of conclusion in discussion.
Inactivated FMDV samples that contain full-length viral RNA carry only a negligible risk of infecting animals. But cattle can be infected by direct injection with naked FMDV RNA, therefore, it should be remarkable these results when diagnosis of FMD.
Response: It is stated in the conclusion of the discussion that inactivated FMDV samples carry only a negligible risk of infecting animals (line 432). Unfortunately, it is not clear to us what the reviewer means by “it should be remarkable these results when diagnosis of FMD”, therefore we were unable to include this statement in the manuscript.
Round 2
Reviewer 2 Report
In my opinion, the changes you made increased the readability and comprehensibility of the text, even for scientists unfamiliar with FMD pathology and histopathology. Thank you.
I only have one very small point, related again with Fig 5. In sub-figure (a) you used arrow for neutrophils infiltrated in the mucosa and arrowhead for degenerating keratinocytes. In sub-figure (d) you reversed them. Do you think it would be possible to harmonize the picture, please?
Author Response
Reviewer 2
In my opinion, the changes you made increased the readability and comprehensibility of the text, even for scientists unfamiliar with FMD pathology and histopathology. Thank you.
I only have one very small point, related again with Fig 5. In sub-figure (a) you used arrow for neutrophils infiltrated in the mucosa and arrowhead for degenerating keratinocytes. In sub-figure (d) you reversed them. Do you think it would be possible to harmonize the picture, please?
Response: Sub-figure (d) has been changed as requested.